# Dynamometer Testing of Energy Efficient Hydraulic Fluids and Fuel Savings Analysis for US Army Construction and Material Handling Equipment

Jill Bramer [1], Eric Sattler [1] and Paul Michael [2,*]

1   US Army DEVCOM Ground Vehicle Systems Center, Force Projection Technology, Warren, MI 48397, USA
2   Fluid Power Institute, Milwaukee School of Engineering, Milwaukee, WI 53202, USA
*   Correspondence: michael@msoe.edu

**Abstract:** The US Army uses MIL-PRF-2104 SAE 10 hydraulic fluid in its construction and material handling equipment. This oil meets basic performance requirements but has not been optimized for hydraulic system efficiency. Hydraulic system efficiency is important because fuel transportation has unique costs and dangers in military applications. Two energy efficient hydraulic fluids were compared to MIL-PRF-2104 SAE 10 in dynamometer testing. The higher efficiency fluids reduced internal leakage flow losses and decreased low-speed motor friction. Fleet-wide fuel savings estimates for US Army construction and material handling machines were derived from engine fuel consumption models, vehicle mission profiles, hydraulic circuit analysis and dynamometer test results. The savings due to reduced fuel consumption was estimated to be $8,000,000 per annum.

**Keywords:** off-highway vehicles; hydraulic efficiency; hydraulic losses; fuel savings

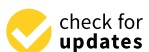



## 1. Introduction

Tribology, the science and technology of interacting surfaces in relative motion, has been identified as an area of vast potential for energy savings. Approximately 23% of the world's total energy consumption originates from tribological contacts. Of that, 20% is used to overcome friction and 3% is used to remanufacture worn parts and spare equipment due to wear and wear-related failures [1]. It is estimated that the potential reductions in energy losses through enhanced tribological engineering in the mining, automotive, papermaking, and fluid power sectors ranges from 11% to 21.5% [2–5]. In the fluid power sector, advanced hydraulic fluids have been identified as an enabling technology for improved efficiency [6].

The US Department of Defense (DOD) consumes more energy than any other federal agency, accounting for 77% of the United States federal government's energy consumption [7]. In FY2019, DOD consumed nearly 3.5 billion gallons of fuel to power ships, aircraft, ground vehicles, and military bases [8]. On a direct cost basis, this amounted to $11 billion in energy expenditures. The largest portion of these expenditures were for jet fuel, at roughly 56% of total DOD energy consumption [9]. For logistic reasons, jet fuel is used in US Army ground vehicles, including commercially sourced US Army construction and material handling machines. Known as Combat Engineering/Material Handling Equipment (CE/MHE) in US Army parlance, construction engineering and material handling equipment perform important functions in support of the US Army's mission. Construction engineering machines build and maintain roads, airfields, helipads, defensive berms, and anti-tank ditches. Material handling equipment load, unload, and transport ISO freight containers, palletized supplies, fuel, and munitions in Army supply distribution operations. Supplying fuel for CE/MHE is costly. It also entails logistical and tactical risks. Hence, reducing fuel demand is a strategic imperative for the US Armed Services [10]. In the early 1970s the US Army adopted a policy of procuring CE/MHE from commercial sources [11]. Commercial OEM fluid property specifications call for the use of modern hydraulic fluid

technology. Hence, these machines are prime candidates for performance enhancement through the use of modern energy efficient hydraulic fluids (EEHF).

US Army Lubrication Orders for field maintenance of construction and material handling equipment specify the use of OE/HDO lubricants. OE/HDO is the U.S. military symbol for MIL-PRF-2104 Lubricating Oil, Internal Combustion Engine, Combat/Tactical Service. The evolution of the MIL-PRF-2104 specification is shown in Table 1. MIL-PRF-2104 oils are intended for use in combat and tactical equipment including the crankcase lubrication of reciprocating compression-ignition engines, heavy-duty automatic and powershift transmissions, hydraulic systems, and non-hypoid gear units. The specification requires that straight- and multi-viscosity grade lubricants fulfill API "C" requirements for compression ignition engines and pass selected Caterpillar and Allison transmission tests. The ability of MIL-PRF-2104 oils to function in engines, powershift transmissions, and hydraulic systems is advantageous in terms of logistics. However, SAE 10 engine oils do not perform as well as a modern hydraulic oil, especially in terms of water tolerance, filterability, air release, and efficiency.

**Table 1.** History of MIL-2104 Lubricants.

| Specification | Year | Civilian Specification | Viscosity Range |
|---|---|---|---|
| Army 2-104 | 1941 | | Monograde heavy-duty engine oil |
| MIL-L-2104 | 1950 | | Monograde heavy-duty engine oil |
| MIL-L-2104A | 1954 | | Monograde heavy-duty engine oil |
| MIL-L-2104B | 1964 | | Monograde heavy-duty engine oil |
| MIL-L-2104C | 1970 | | Monograde heavy-duty engine oil |
| MIL-L-2104D | 1983 | API CD, SE. DD Allison C-3 | SAE 10, 30, 40, and 15W/40 |
| MIL-L-2104E | 1988 | API CD, CF-2, DDA C-3, Cat TO-2 | SAE 10, 30, 40, and 15W/40 |
| MIL-L-2104F | 1992 | API CD-II, ATD C-4, Cat TO-4 | SAE 10, 30, 40, and 15W/40 |
| MIL-PRF-2104G | 1997 | API CF-4/CG, ATD C-4, Cat TO-4 | SAE 10, 30, 40, and 15W/40 |
| MIL-PRF-2104H | 2004 | API CI-4, ATD C-4, CAT TO-4 | SAE 40, 15W/40, 5W/40 |
| MIL-PRF-2104J | 2014 | API CH-4/CI-4, ATD C-4, CAT TO-4 | SAE 10, 30, 40, and 15W/40 |
| MIL-PRF-2104K | 2016 | API CH-4/CI-4, ATD C-4, CAT TO-4 | SAE 40, 15W/40 and SCPL |
| MIL-PRF-2104L | 2017 | API CH-4/CI-4, ATD C-4, CAT TO-4 | SAE 10, 30, 40, 15W/40 and SCPL |
| MIL-PRF-2104M | 2017 | API CH-4/CI-4, ATD C-4, CAT TO-4 | SAE 10, 30, 40, 15W/40 and SCPL |
| MIL-PRF-32626 | 2019 | Establish MIL spec for SMPL | SAE 0W/20 |
| MIL-PRF-2104N | 2021 | API CH-4/CI-4, ATD C-4, CAT TO-4 | SAE 10, 30, 40, and 15W/40 |

In a typical fluid power system, the mechanical energy of an electric motor or internal combustion engine is transferred to the fluid medium by a positive displacement pump and the controlled motion of the fluid is used to actuate cylinders, motors, and other machine components. Flow and pressure losses due to internal leakage, viscous drag, friction, and fluid compressibility reduce the efficiency of fluid power systems [12]. Efficiencies in the context of fluid power are determined from the ratios of actual to theoretical flow, torque, and power outputs (division). Losses are determined from the differences between the theoretical and actual outputs (subtraction) [13]. The efficiency of power transmission by hydraulic machinery can be affected by properties of the fluid. Shear-stable high viscosity index hydraulic fluids have been found to improve the volumetric efficiency of hydraulic pumps by reducing internal flow losses, as have high bulk modulus fluids [14,15]. Flow losses generate heat and reduce the amount of flow available to actuate hydraulic cylinders, which negatively impacts machine productivity. Hydraulic fluids that have a low traction coefficient have been found to increase the mechanical efficiency of hydraulic motors by reducing low-speed friction [16]. The frictional losses that occur in motors generate heat and reduce the torque available to move the payload at low speeds. Increasing the torque output of a hydraulic motor and the effective flow output of a hydraulic pump decreases the work-specific fuel consumption of the machine. The potential of energy efficient hydraulic fluids to reduce the work-specific energy consumption of CE/MHE is investigated below. This investigation includes a combination of dynamometer testing and

fuel consumption models to estimate the potential fleet-wide fuel savings for 11 US Army CE/MHE vehicle platforms.

## 2. Materials and Methods

### 2.1. Test Fluids

Table 2 lists the three hydraulic fluids that were evaluated in this study. The SAE 10 fluid met the MIL-PRF-2104M specification and was procured from a supplier on the qualified products list (QPL). Fluids "A" and "B" were commercial EEHF produced by global suppliers. All three fluids met the ISO 46VG specification and had a relatively high viscosity index. Polymethacrylate viscosity index improvers were used in the energy efficient fluid formulations. The high viscosity index of the SAE 10 engine oil was attributed to dispersant polymer additives that are commonly used in crankcase oil detergent-inhibitor packages. All three oils had good shear-stability as measured in the ASTM D5621 Sonic Shear Test.

**Table 2.** Properties of test fluids. (Colors correspond to those used in reporting the dynamometer test results).

| Fluid | Test | SAE 10 | EEHF-A | EEHF-B |
|---|---|---|---|---|
| Viscosity at 40 °C, cSt | ASTM D445 | 45.47 | 46.52 | 45.83 |
| Viscosity at 100 °C, cSt | ASTM D445 | 7.64 | 8.58 | 8.36 |
| Viscosity Index | ASTM D2270 | 136 | 164 | 160 |
| Shear Stability, % vis loss at 40 °C | ASTM D5621 | 3.6 | 3.2 | 3.7 |
| Density at 15 °C, g/mL | ASTM D4052 | 0.8623 | 0.8327 | 0.8516 |

### 2.2. Hydraulic Dynamometer

Hydraulic system flow and motor torque losses were evaluated in the hydraulic circuit shown in Figure 1. The circuit incorporated an open-loop variable-displacement axial piston pump. The pump inlet temperature was controlled to 50 °C or 80 °C (±1 °C). The pump angular velocity was adjusted to 1200 rpm or 1800 rpm depending upon hydraulic motor input flow requirements. Pump displacement was controlled by a proportional electrohydraulic valve that adjusted the swash plate angle to maintain a desired pump outlet pressure. The pump supplied fluid power to the test motors to yield rotational frequencies ranging from 1 to 1400 rpm.

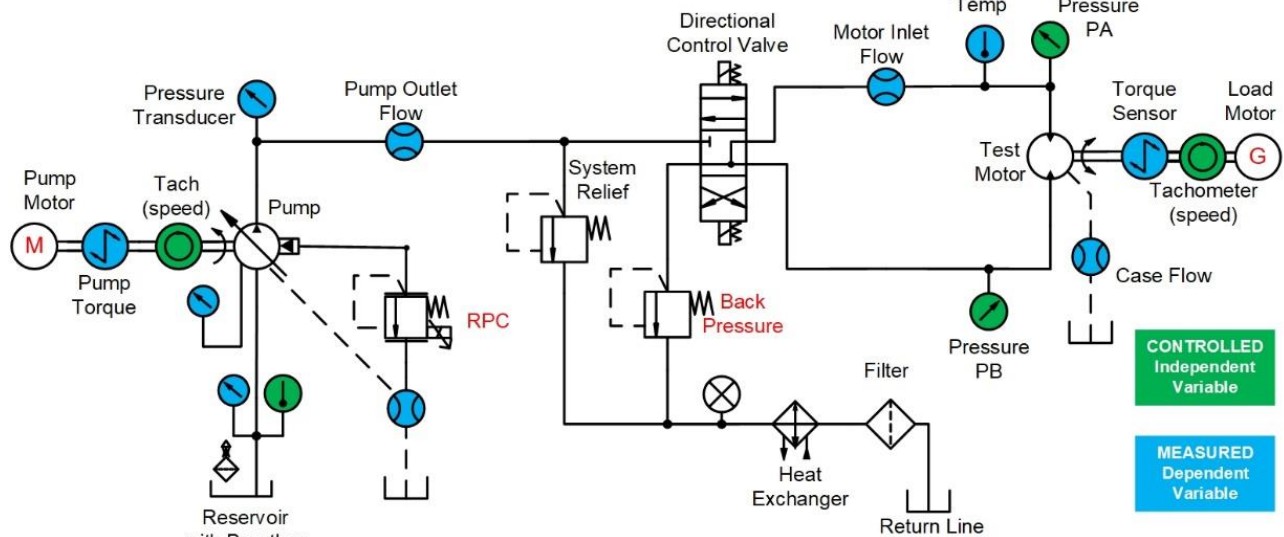

**Figure 1.** The dynamometer included a pressure-compensated variable displacement piston pump and a hydraulic motor.

The effect of fluid selection on torque and flow losses was evaluated for the motors listed in Table 3. These specific units were selected because they are representative of motors used in CE/MHE vehicles. Low-speed data was collected using a modified ISO 4392-1 procedure. [17] Each motor and fluid combination was evaluated seven times at seven pressures between 69 and 276 Bar. These test results were used to establish the 95% confidence interval of the mean low-speed torque losses. High-speed data was collected for each motor and fluid combination using a modified ISO 4409 procedure. [18] Latin Hyperspace (LHS) sampling was used to select the high-speed test points. LHS is a pseudo-randomized sampling methodology that has been found to produce higher fidelity torque and flow models than orthogonal methods [19]. Each motor and fluid combination was evaluated four times at high speeds. These test results were used to compare the input and output power requirements of the fluids. The sample plan for the radial piston motor is shown in Figure 2: Test plan for radial piston motor showing ISO 4392-1 low-speed (blue) and ISO 4409 high-speed (red) test points. Sample selection for the axial and variable piston motors was similar but had a higher upper bound for speed.

**Table 3.** Test motor specifications.

| Motor Type | Radial Piston | Axial Piston | Variable Axial Piston |
|---|---|---|---|
| Displacement, cc/rev | 213 | 100 | 45.2/135.6 |
| Rated speed, RPM max | 570 | 3300 | 3200 |
| Rated pressure, Bar max | 400 | 420 | 450 |
| Mass, kg | 26 | 34 | 56 |
| Displacement/weight ratio, cc/kg | 8.2 | 2.9 | 2.4 |

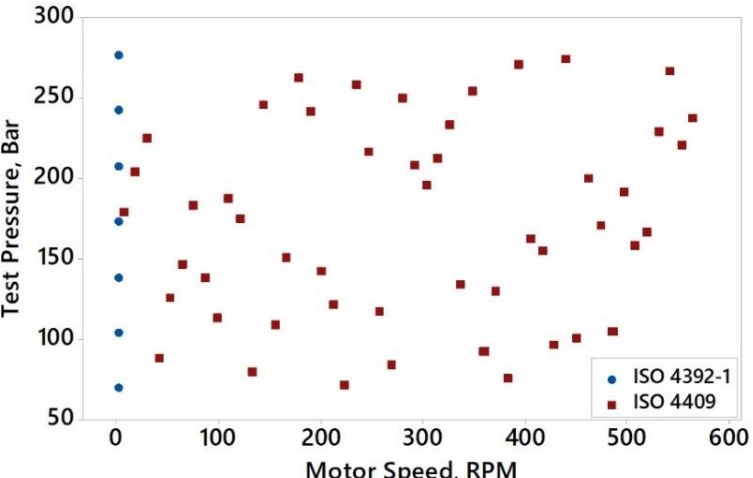

**Figure 2.** Test plan for radial piston motor showing ISO 4392-1 low-speed (**blue**) and ISO 4409 high-speed (**red**) test points.

## 3. Results

### 3.1. Hydraulic Motor Torque Losses

The torque output of each hydraulic motor was measured for the fluids at 1RPM per ISO 4392-1. Fluid effects were compared using the two-sample t-test at a 95% Confidence Interval ($\alpha = 0.05$). Since the torque loss measurements were highly repeatable, the confidence interval bars are very narrow and approximately the same size as the mean symbol. Figure 3 shows the 95% CI of the mean torque losses for the radial piston motor. The difference between the results at 50 °C and 80 °C were not statistically significant. Thus, results at both temperatures were combined in the figure. As the motor differential pressure increased, the torque losses for the fluids also increased. This is the result of higher contact pressures at the tribological interfaces. At maximum pressure, the mean torque loss for

fluid "A" was 28.1% lower than SAE 10, while the torque loss for fluid "B" was 18.3% lower. Lower torque losses are the result of a reduction in friction. Reducing motor friction at elevated pressures when a machine is operating at high intensity improves productivity because more torque is available to propel the vehicle and move the payload.

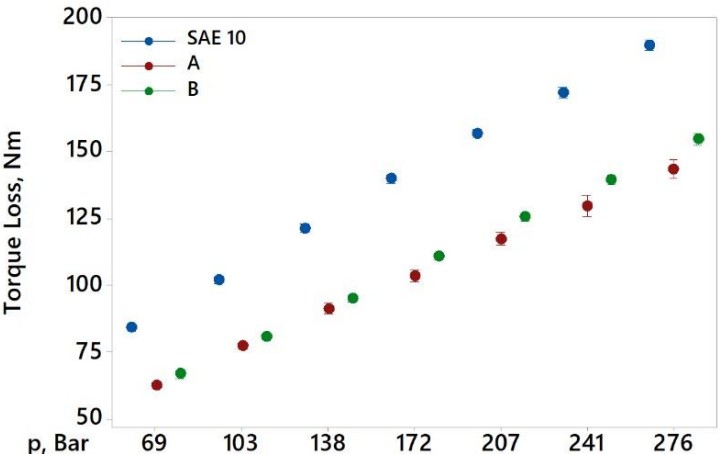

**Figure 3.** Mean and 95% CI for the radial piston motor torque losses at 1RPM. Results for 50 °C and 80 °C are combined.

Figure 4 shows the torque losses for the three motors at the maximum test pressure. The axial and variable motors exhibited lower torque losses than the radial motor because their volume displacement per revolution was lower. At maximum pressure fluid "A" reduced the torque losses by 26.8% in the axial piston motor and 16.2% in the variable motor. Fluid "B" was less effective at reducing friction. It decreased torque losses 8.6% in the axial piston motor and 6.6% in the variable displacement axial piston motor. The motors differ in response to oil chemistry because they have different tribological elements, materials of construction, and contact mechanics. In spite of these differences, the energy efficient hydraulic fluids increased the output of all three motors under low-speed high-torque conditions.

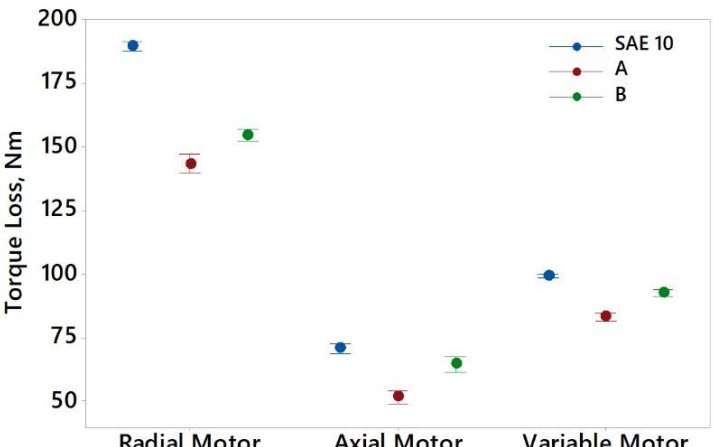

**Figure 4.** Mean and 95% CI for the motor torque losses at 276 BAR and 1 RPM.

The high-speed torque losses for the variable displacement axial piston motor are shown in Figure 5. Higher torque losses for the SAE10 oil were observed at motor speeds less than 85 RPM. Above this speed, motor lubrication transitions from boundary and mixed film regimes to hydrodynamic lubrication. Above 100 RPM the differences in torque losses were insignificant. A discontinuity in the torque loss plot can be observed at 500 RPM. This is the result of the variable displacement motor shifting to a lower displacement,

thereby reducing torque losses at high speeds. Likewise reducing low-speed torque losses did not increase the high-speed torque losses in the radial and axial piston motor. Hence improvements in low-speed performance do not result in a compromise to the high-speed performance of the motors.

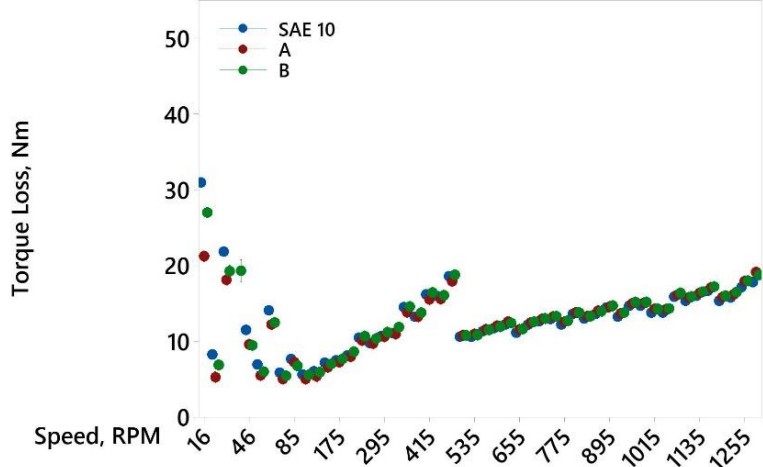

**Figure 5.** Mean and 95% CI for the variable displacement motor torque losses at 50 °C and high speeds.

### 3.2. Hydraulic System Flow Losses

The flow losses for each motor and fluid combination at 50 °C and 80 °C were evaluated per ISO 4409. Flow losses are the sum of the pump, pressure compensator, directional control valve, and motor internal leakage flows. Controlling internal leakage is important because heat is produced when a hydraulic fluid depressurizes as it passes across a tribological gap. This results in an increase in the energy required to operate hydraulic heat exchangers. In addition, the resulting flow loss reduces the amount of fluid available to actuate the hydraulic system. Figure 6 shows the system flow losses for the axial piston motor at 80 °C. As expected, flow losses increased with pressure. The increase was from 15 LPM at 69 BAR to 45 LPM at 276 Bar.

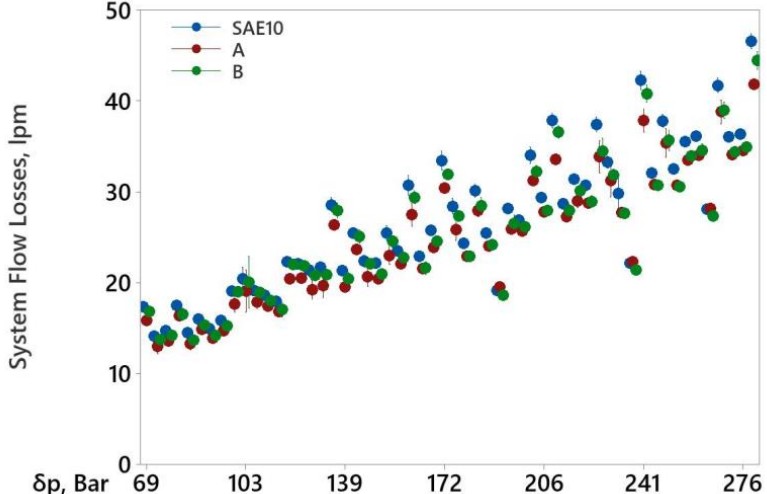

**Figure 6.** Mean and 95% CI for system flow losses at 80 °C for the axial piston motor. Note that internal leakage flow increases with differential pressure (δp).

Figure 7 shows the percent change in system flow losses relative to SAE 10 for each motor and fluid combination at a fluid temperature of 80 °C. On average, Fluid A reduced

the flow losses by 5.2% to 7.7% relative to SAE 10. Fluid B reduced the flow losses by 3.8% to 5.3% relative to SAE 10. These results are consistent with previous reports where the flow loss reductions afforded by shear-stable multigrade hydraulic fluids ranged from 4 to 6%. [14]

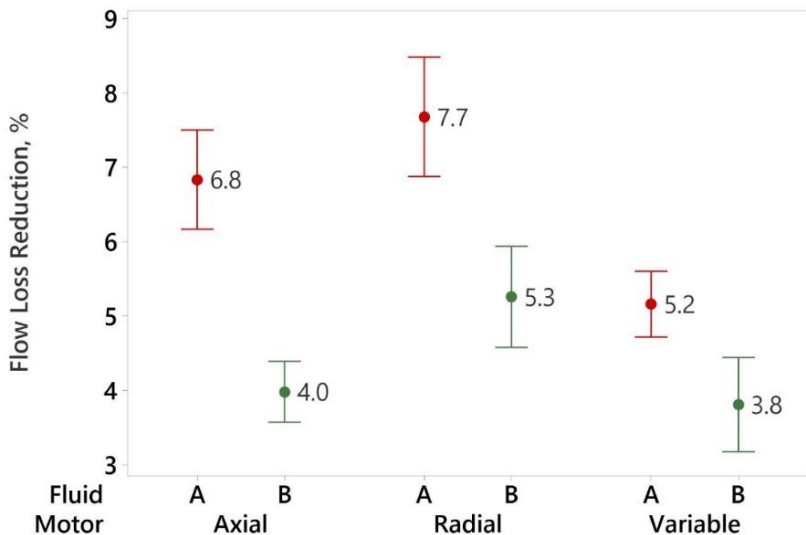

**Figure 7.** Percent Flow Loss Reduction Relative to SAE 10 at 80 °C.

### 3.3. Pump Input and Motor Output Power

ISO 4391 defines pump input power as the mathematical product of shaft speed and torque. As shown in Figure 7 above, the use of EEHF decreased the system flow losses. A reduction in flow losses causes the swashplate of a pressure compensated variable displacement pump to de-stroke. De-stroking the pump reduces the torque required to produce pump rotation, thereby reducing the input power to the hydraulic system.

ISO 4391 defines motor output power as the mathematical product of shaft speed and torque. As shown in Figure 4 above, EEHF can decrease low-speed torque losses in motors. This reduction in torque losses increases the power output of the motor. Hence EEHF are capable of decreasing pump input power and increasing motor output power in hydraulic equipment.

The extent to which pump input and motor output power are affected by fluid properties depends upon the operating conditions and the tribological characteristics of the hydraulic components. In order to visualize the relative input and output power effects, the results were normalized in Figures 8–10, where the ratios of the input and output power of the test fluids are compared with SAE 10 for each motor. SAE 10 requires more input power to produce a given level of output power than the EEHF for points below the diagonal line, while the EEHF requires more input power to produce a given level of output power than the SAE 10 for points above the diagonal line. At the circled point in Figure 8, the axial piston motor produced 6% more torque with 2% less pump input power relative to SAE 10. As shown in Figure 9 the EEHF also improved the power ratios in the radial piston motor. Relative to the axial piston motor, the response of the radial piston motor was more pronounced. It is hypothesized that this is due to the higher displacement-to-weight ratio and lower operating speed of the radial piston motor. (See Table 3) This combination produces high contact pressures and low sliding speeds that shift tribological conditions toward the boundary lubrication regime where surface active additives can significantly impact friction. It is noteworthy that relative to SAE 10, EEHF-A produced more output power with less input power in the radial piston motor for all conditions of pressure and speed at 80 °C.

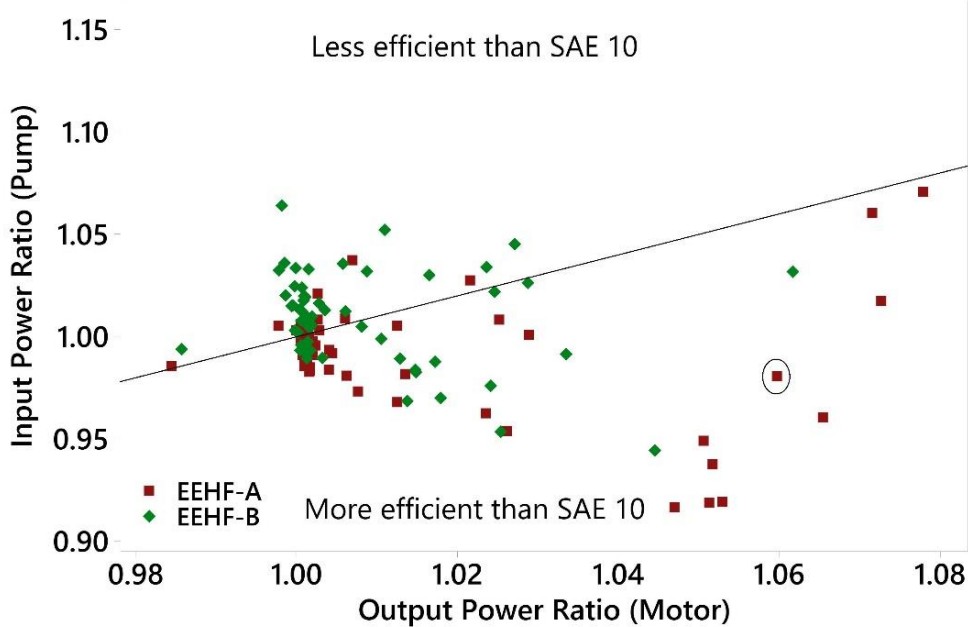

**Figure 8.** Input and output power ratio for axial piston motor at 80 °C. (At the circled point in Figure 8, the axial piston motor produced 6% more torque with 2% less pump input power relative to SAE 10).

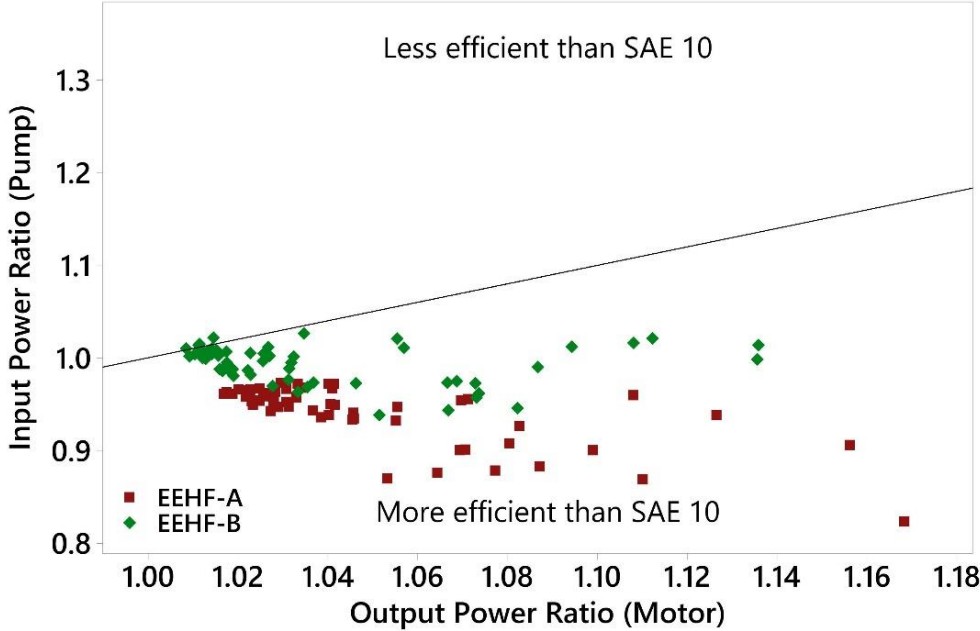

**Figure 9.** Input and output power ratio for radial piston motor at 80 °C.

The effect of EEHF on the input and output power of the variable displacement axial piston motor was different. As shown in Figure 10, many of the points are above the diagonal line, which means the EEHF produced less motor output power for a given level of pump input power relative to SAE 10. As shown in Table 3, the displacement-to-weight ratios of the fixed displacement and variable displacement axial piston motors are comparable. It is hypothesized that the metallurgy of the tribological elements in the variable displacement motor were relatively inert and less susceptible to the effects of surface active additives in the fluid.

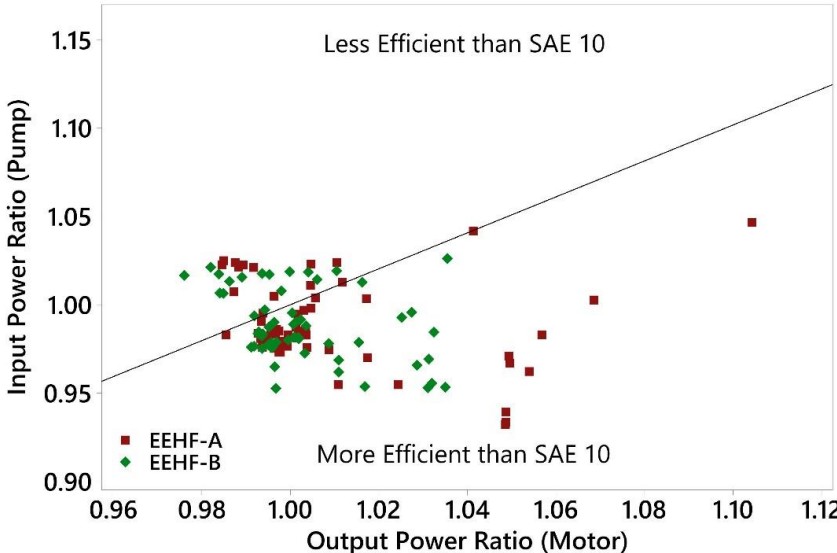

**Figure 10.** Input and output power ratio for the variable displacement axial piston motor at 80 °C.

*3.4. Normalized Work-Specific Efficency*

Normalized work-specific efficiency is useful for comparing fluids because hydraulic system performance is highly affected by operating conditions. It is determined by dividing the hydraulic system efficiency of the test fluid at a given operating speed, pressure, and temperature, by the hydraulic system efficiency of a baseline fluid under identical conditions. In this case, EEHF A and B are the test fluids, and SAE 10 is the baseline. Hydraulic system efficiency is defined as the ratio of motor output power to pump input power. When this ratio is higher for the EEHF than SAE 10, the normalized work-specific efficiency is greater than 1.0.

Figures 11 and 12 show normalized work-specific system-level efficiencies measured during testing of the axial and radial piston motors. EEHF A improved system efficiency for both motors by at least 10% at 1 RPM. (Multiple data points are plotted on the origin of the X-axis because 1 RPM tests were conducted for seven different pressure levels.) Interestingly, EEHF A exhibits two distinct bands of results in radial piston motor testing. The upper red symbols represent data collected at 80 °C. The lower red symbols represent data collected at 50 °C. This effect persists throughout the hydrodynamic range. It is hypothesized that this difference is due to the lower viscosity of SAE 10 at 80 °C.

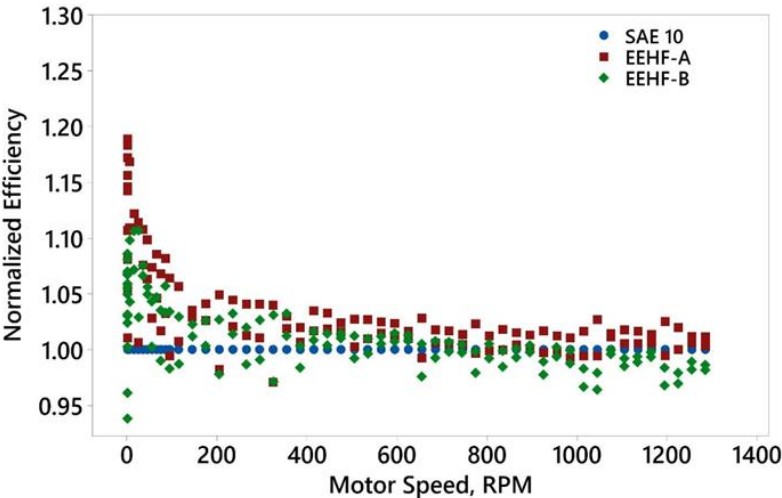

**Figure 11.** Normalized work-specific efficiency of the axial piston motor at 50 °C and 80 °C relative to SAE 10.

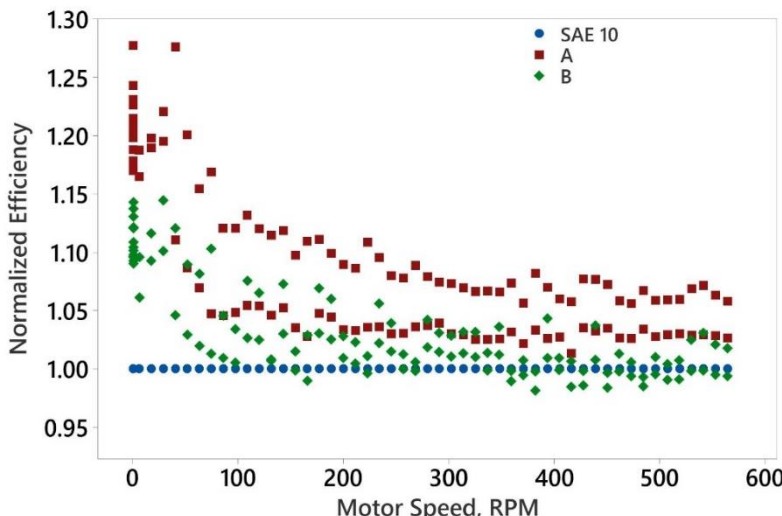

**Figure 12.** Normalized work-specific efficiency of the radial piston motor at 50 °C and 80 °C relative to SAE 10.

As shown in Figure 13, the variable displacement motor exhibited less benefit from the use of energy efficient fluids. While both EEHF performed better than SAE 10 at low speeds, SAE 10 was consistently more efficient than Fluid A at higher speeds. Fluid B was roughly equivalent. Thus it must be acknowledged that the impact of EEHF on hydraulic system efficiency is affected by hydraulic component design and operating conditions.

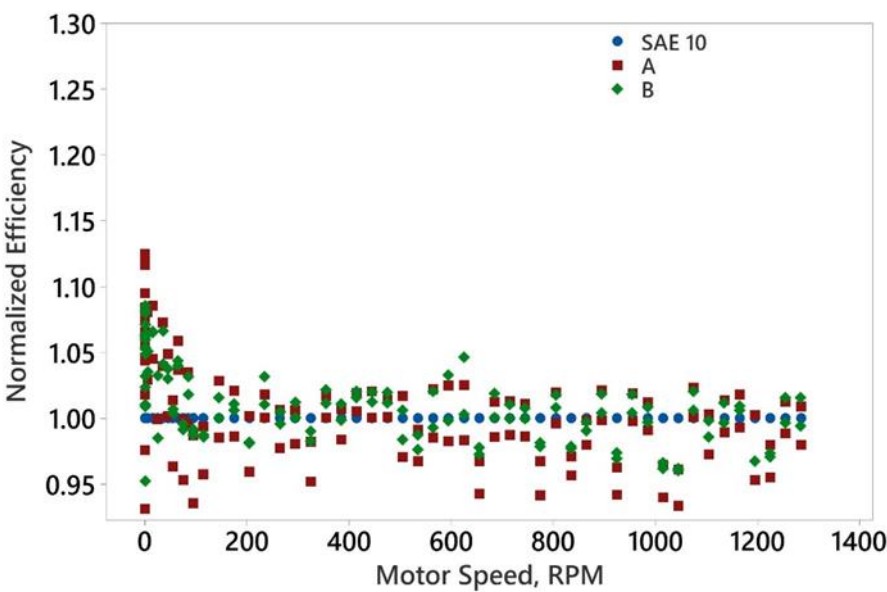

**Figure 13.** Normalized efficiency of the variable displacement axial piston motor at 50 °C and 80 °C.

From an equipment manufacturer's standpoint, improving the low-speed hydraulic performance is desirable because the size of the hydraulic power unit is based on the amount of torque required to accelerate a fully loaded machine from a stopped position. Hence, improved efficiency at low speeds can facilitate the use of smaller and less energy intensive engines and pumps. In field application, low-speed, high-pressure efficiency improvements enhance productivity when the machine is working hardest, such as when it is driving into a gravel pile, swinging a bucket of soil out of a trench, or loading a pallet of supplies into an ISO container. Improvements in efficiency under these conditions enable these tasks to be performed more quickly, which is particularly desirable when conducting high-risk operations. Reductions in leakage flow rates also have a beneficial effect on

vehicle productivity because more flow is available to power machine actuators. There also is a fuel consumption impact. When leakage flows are reduced, hydraulic pumps de-stroke, which reduces the output torque required of the engine. It also reduces the thermal loading of hydraulic heat exchangers, decreasing the amount of energy required of the engine to control the hydraulic system temperature. Hence the net effect of EEHF is to reduced fuel consumption and increase machine productivity.

### 3.5. Fuel Consumption Analysis

In off highway vehicles, work-specific fuel consumption rather than miles per liter is used to characterize fuel efficiency. An analysis of work-specific fuel consumption combines measurements of fuel rate (liters/hour) and machine productivity (hours/ton) to yield fuel consumption measurements in units of (liters fuel/ton). As one would imagine, work-specific fuel consumption is highly dependent upon the size and speed of the engine, hydraulic system architecture, and task intensity. An analysis was conducted to assess the potential fuel savings and economic impact of EEHF.

The engine power was determined for each machine using information published by the manufacturer. The hydraulic power was determined for each machine based on an analysis of the circuit schematics, system components, and published specifications for maximum pressure and flow rates. In cases where the theoretical hydraulic power was greater than the engine power, a ratio of 1.0 was assumed. Most of the machines had a high ratio of hydraulic to engine power as shown in Table 4.

**Table 4.** Ratio of hydraulic to engine power.

| Machine | Engine | Size (L) | Speed (RPM) | Engine Power (kw) | Hydraulic Power (kw) | Hydraulic/Engine Power Ratio |
|---|---|---|---|---|---|---|
| 120M Grader | C6.6 ACERT | 6.6 | 2150 | 103 | 103 | 1.00 |
| 400 SSL | IVECO F5C | 3.2 | 2500 | 69 | 69 | 1.00 |
| 580M BHL | CASE 445T/M2 | 4.5 | 2200 | 67 | 67 | 1.00 |
| 621B Scraper | C15 ACERT | 15.2 | 1800 | 272 | 183 | 0.67 |
| 924H Loader | C6.6 ACERT | 6.6 | 2300 | 96 | 96 | 1.00 |
| 966H Loader | C11 ACERT | 11.1 | 1800 | 195 | 195 | 1.00 |
| Atlas II 10K | Deere | 4.5 | 2400 | 129 | 127 | 0.98 |
| D6K Dozer | C6.6 ACERT | 6.6 | 2100 | 93 | 93 | 1.00 |
| D7R-II Dozer | CAT 3176C | 10.3 | 2100 | 179 | 179 | 1.00 |
| JD240 HYEX | PowerTech 6.8 | 6.8 | 2000 | 132 | 132 | 1.00 |
| RTCH | Cummins QSM 11 | 10.8 | 2100 | 298 | 217 | 0.73 |

Hourly fuel consumption rates for each machine were estimated from an American Society of Agricultural and Biological Engineers (ASABE) tractor model [20], an EPA heavy duty diesel vehicle model [21], and tables published by Caterpillar [22]. The ASABE model is based upon hundreds of fuel consumption tests conducted on tractors at the Nebraska Tractor Test Laboratory. In the ASABE model, fuel consumption rates are determined from the ratio of equivalent PTO power to rated PTO power. This method has been adapted to CE/MHE machines by equating PTO power to the hydraulic/engine power ratio shown in Equation (1).

$$FR = \left( 0.22 \left( \frac{P_{HYD}}{P_{ENG}} \right) + 0.096 \right) \times P_{HYD} \tag{1}$$

In Equation (1), *FR* is the fuel consumption rate in liters per hour, $\frac{P_{HYD}}{P_{ENG}}$ is the decimal ratio of the hydraulic to engine power, and $P_{HYD}$ is the maximum hydraulic power in kW.

"MOVES" is an EPA simulation tool for estimating vehicle emissions. Models are available for a wide range of vehicles. The model for heavy duty diesel engines is based upon construction equipment, over-the-road machines, and transit busses. As shown in

Equation (2), fuel consumption rates are determined from engine speed, displacement, mean effective pressure and the lower heating value of the fuel.

$$FR = \left( \frac{\frac{N \times V}{2000}}{LHV \times \eta} \right) \times (fmep + bmep) \tag{2}$$

In Equation (2), $N$ is the engine speed, rpm, $V$ is the engine displacement, litres, $LHV$ is the loweer heating value of fuel, kj/kg, $fmep$ is friction mean effective pressure, $bmep$ is brake mean effective pressure and, $\eta$ is the engine efficiency.

As shown in Table 5, the fuel consumption rates for the EPA model were lower than the rates predicted by the ASABE model. This is because the ASABE model was developed for high tractive effort agricultural machines while the EPA model was developed for on-road vehicles. Since Caterpillar's data generally fell between the two values, an average of the EPA and ASABE models was used to estimate the fuel consumption rate for machines that did not have published fuel rates from Caterpillar.

**Table 5.** Vehicle hourly fuel consumption estimates.

| Machine | Engine | Caterpillar Fuel Rate (Liters/h) | ASABE Fuel Rate (Liters/h) | EPA Fuel Rate (Liters/h) | Estimated Fuel Rate (Liters/h) |
|---|---|---|---|---|---|
| 120M Grader | C6.6 ACERT | 22.3 | 32.5 | 14.0 | 22.3 |
| 400 SSL | IVECO F5C | | 21.8 | 7.9 | 14.9 |
| 580M BHL | CASE 445T/M2 | | 21.2 | 9.8 | 15.5 |
| 621B Scraper | C15 ACERT | | 44.7 | 27.1 | 35.9 |
| 924H Loader | C6.6 ACERT | 15.0 | 30.3 | 15.0 | 15.0 |
| 966H Loader | C11 ACERT | 20.5 | 61.6 | 19.8 | 20.5 |
| Atlas II 10K | Deere | | 39.7 | 10.7 | 25.2 |
| D6K Dozer | C6.6 ACERT | 26.4 | 29.2 | 13.7 | 26.4 |
| D7R-II Dozer | CAT 3176C | 39.0 | 56.6 | 21.4 | 39.0 |
| JD240 HYEX | PowerTech 6.8 | | 41.7 | 13.5 | 27.6 |
| RTCH | Cummins QSM 11 | | 55.6 | 22.4 | 39.0 |

The total annual fuel consumption of CE/MHE was determined from fleet size, hourly fuel consumption rates, and annual hours of operation data. US Army Operational Mission Statements and Mission Profile (OMS/MP) specifications describe how a vehicle will be used in future active deployment. These specifications were used to estimate the annual service hours for each platform. As shown in Table 6, the Atlas II 10K OMS/MP specification indicated that it could operate 13.8 h per day or nearly 414 h per month, which equates to 4968 h per annum. This represents two to three times typical commercial use of such vehicles. The OMS/MP operating hours for the graders, scrapers, and loaders were also very high. Pre-positioning and storage of military equipment in strategic areas is an effective deterrent and consequently a sizeable portion of the CE/MHE fleet is inactive. As shown in Table 7, it is assumed that half of the CE/MHE fleet is inactive, which reduces to total operating hours to a more realistic level.

**Table 6.** Mission profile for the Atlas II 10K forklift indicating Light Duty (LD) and Heavy Duty (HD) working hours.

| Task Element | Work Intensity | Daily Shift (h) | 30 Day Total (h) |
|---|---|---|---|
| Start-up procedures | LD | 0.3 | 9 |
| Movement within site | HD | 3 | 90 |
| Load, stuff, move, unload, and unstuff | HD | 7 | 210 |
| Movement between sites | LD | 2.75 | 82.5 |
| End of shift procedures | LD | 0.75 | 22.5 |
| TOTALS | | 13.8 | 414 |

**Table 7.** Annual Fuel Consumption Estimates.

| Machine | Fleet Size | Deployment Level, % | Estimated Fuel Rate (Liters/h) | Lite Duty (h/year) | Heavy Duty (h/year) | Fuel Consumption (Liters/year) |
|---|---|---|---|---|---|---|
| 120M Grader | 750 | 50 | 22.3 | 720 | 5040 | 44,555,400 |
| 400 SSL | 2020 | 50 | 14.9 | 325 | 2640 | 41,685,730 |
| 580M BHL | 645 | 50 | 15.5 | 325 | 2640 | 13,846,538 |
| 621B Scraper | 575 | 50 | 35.9 | 720 | 5040 | 54,991,620 |
| 924H Loader | 320 | 50 | 15.0 | 720 | 5040 | 12,787,200 |
| 966H Loader | 260 | 50 | 20.5 | 720 | 5040 | 14,199,120 |
| Atlas II 10K | 4546 | 50 | 25.2 | 378 | 4590 | 271,574,040 |
| D6K Dozer | 175 | 50 | 26.4 | 720 | 2880 | 7,318,080 |
| D7R-II Dozer | 1295 | 50 | 39.0 | 720 | 2880 | 79,999,920 |
| JD240 HYEX | 265 | 50 | 27.6 | 720 | 2042 | 8,520,810 |
| RTCH | 878 | 50 | 39.0 | 378 | 3222 | 57,752,557 |
| Total | 11,729 | | | | Liters/year | 607,231,014 |

Elements of each machine's OMS/MP were classified in terms of light (LD) or heavy duty (HD) work intensity. A comparison of fuel consumption rates under light and heavy-duty intensity levels indicates that a 40% difference is typical. This ratio was used to determine the annual hourly fuel consumption rate during low-intensity operations as shown in Equation (3). Based upon these assumptions, CE/MHE fuel consumption exceeds 600 million liters per year at a 50% deployment level as shown in Table 7.

$$Fuel\ consumption = (fleet\ size \times DL)[(Hrs_{HD} \times FR) + (Hrs_{LD} \times FR \times 0.4)] \quad (3)$$

The reduction in fuel consumption for each machine was estimated based upon an analysis of the vehicle kinematics, hydraulic circuit, and dynamometer testing of SAE10 and EEHF. In addition to the hydraulic motor tests reported above, dynamometer testing was conducted in gear and electrohydraulic axial piston pumps (unpublished results). As shown in Table 8, the projected efficiency gain in hydrostatic circuits, vane pumps and hydraulic motors was 3%. The projected efficiency gain was 2% in piston pumps and 1.5% in gear pumps. Based upon a partitioning of the hydraulic power distribution (Hydraulic Energy (%)), the net hydraulic efficiency gain was determined for each machine.

**Table 8.** Estimated Hydraulic Efficiency Improvement.

| Machine | | Function | Pump/Motor Type | Hydraulic (HP) | Hydraulic Energy | Efficiency Gain |
|---|---|---|---|---|---|---|
| 120M Grader | 1 | Implement/Steer | Piston pump | 111.82 | 50% | 2.0% |
| | 2 | Fan/Brake | Piston pump | 26.94 | 8% | 2.0% |
| | 3 | Front wheel drive | Hydrostatic (2) | 104.46 | 40% | 3.0% |
| | 4 | Charge pump L/R | Fixed displ (2) | 2.27 | 2% | 0.0% |
| 400 Skid Steer Loader | 1 | Propulsion | Hydrostatic (2) | 78.76 | 70% | 3.0% |
| | 2 | Implement | Gear pump | 40.26 | 25% | 1.5% |
| | 3 | Charge pump | Gear pump | 0.00 | 5% | 1.5% |
| 580M Backhoe Loader | 1 | Loader | Tandem gear | 50.71 | 50% | 1.5% |
| | 2 | Backhoe & Steering | Tandem gear | 67.62 | 50% | 1.5% |
| 621G Scraper C9 | 1 | Steering | Vane pump | 80.08 | 50% | 3.0% |
| | 2 | Implement | Vane pump | 103.29 | 50% | 3.0% |
| 924H Wheel Loader | 1 | Implement | Piston Pump | 86.78 | 60% | 2.0% |
| | 2 | Steering | Piston Pump | 48.48 | 30% | 2.0% |
| | 3 | Fan & Brake | 288–4162 | 8.46 | 5% | 1.5% |
| | 4 | Fan drive | Piston motor | — | 5% | 3.0% |

**Table 8.** *Cont.*

| Machine | | Function | Pump/Motor Type | Hydraulic (HP) | Hydraulic Energy | Efficiency Gain |
|---|---|---|---|---|---|---|
| 966H Wheel Loader | 1 | Steering | Piston pump | 94.78 | 60% | 2.0% |
| | 2 | Implement | Piston pump | 221.25 | 30% | 2.0% |
| | 3 | Fan & Brake | Piston pump | 32.55 | 5% | 2.0% |
| | 4 | Fan drive | Piston motor | 0.00 | 5% | 3.0% |
| Altas II 10K Fork Lift | 1 | Steering & brake | Tandem gear | 96.27 | 65% | 3.0% |
| | 2 | Boom | Tandem gear | 14.70 | 5% | 3.0% |
| | 3 | Fork cylinders | Piston Pump | 55.15 | 30% | 2.0% |
| D6K Dozer | 1 | Implement | Piston pump | 69.73 | 20% | 2.0% |
| | 2 | Propel L/R | Hydrostatic (2) | 92.64 | 70% | 3.0% |
| | 4 | Winch | Piston pump | 99.40 | 3% | 2.0% |
| | 5 | Winch drive | Piston motor | 1.74 | 1% | 3.0% |
| | 6 | Fan | Gear pump | 27.54 | 3% | 1.5% |
| | 7 | Fan drive | Gear motor | — | 1% | 3.0% |
| | 8 | Charge pumps | Gear pumps (3) | 0.30 | 5% | 1.5% |
| D7R-II Dozer | 1 | Blade and steering | Piston pump | 250.47 | 100% | 2.0% |
| JD240 Excavator | 1 | Main power | Bent axis pumps (2) | 172.00 | 35% | 0.0% |
| | 2 | Track drive L/R | Piston motors (2) | — | 35% | 3.0% |
| | 3 | Swing | Piston motor | 3.01 | 20% | 3.0% |
| | 4 | Control system | Gear pump | 172.00 | 10% | 1.5% |
| Rough Terrain Cargo Handler | 1 | Steering | Tandem Piston | 76.04 | 35% | 2.0% |
| | 2 | Steering | Tandem Piston | 76.04 | 35% | 2.0% |
| | 3 | Top handler | Piston Pump | 54.90 | 20% | 2.0% |
| | 4 | Boom/brake | Vane pump | 7.00 | 5% | 3.0% |
| | 5 | Auxiliary systems | Fixed displacement | 2.02 | 3% | 0.0% |
| | 6 | Cooling fan | Fixed displacement | 1.28 | 2% | 0.0% |

The estimated hydraulic efficiency gains from the use of EEHF ranged from 1.5% in the 580M backhoe loader to 3.0% in the 621G scraper. Based upon results reported in the literature for commercial EEHF, these projected hydraulic efficiency gains are conservative.

The potential savings afforded by EEHF was calculated based upon the annual fuel consumption estimate in Table 7 and the efficiency gain estimate in Table 8. The annual volume was multiplied by the vehicle efficiency improvement to calculate the volume of fuel saved. In turn, the fuel volume was multiplied by the cost per liter. The Defense Logistics Agency (DLA) costs listed in Table 9 were the direct costs paid for fuel during the study period. The In-Theatre costs represent the estimated fully burdened cost of fuel delivered to active US military operations. These values are much higher than the direct cost because fuel is transported to remote military bases via air cargo. Based upon the preceding analysis, the annual fuel savings afforded by the use of EEHF ranges from $8 million to $64 million depending upon the cost basis.

**Table 9.** Annual fuel savings based upon Defense Logistics Agency Direct and Fully Burdened Costs.

| Vehicle Platform | Hydraulic Efficiency Improvement | Vehicle Efficiency Improvement | DLA-Energy LOW $0.61/Liter | DLA-Energy HIGH $1.04/Liter | In-Theatre LOW $3.73/Liter | In-Theatre HIGH $4.61/Liter |
|---|---|---|---|---|---|---|
| 120M Grader | 2.40% | 2.40% | $649,998 | $1,097,218 | $3,924,985 | $4,844,426 |
| 400 SSL | 2.60% | 2.60% | $655,313 | $1,106,191 | $3,957,083 | $4,884,043 |
| 580M BHL | 1.50% | 1.50% | $128,254 | $216,498 | $774,458 | $955,877 |
| 621G Scraper | 3.00% | 2.00% | $685,401 | $1,156,981 | $4,138,769 | $5,108,290 |
| 924H Loader | 2.00% | 2.00% | $160,067 | $270,198 | $966,554 | $1,192,973 |
| 966H Loader | 2.10% | 2.10% | $179,935 | $303,736 | $1,086,528 | $1,341,051 |

**Table 9.** *Cont.*

| Vehicle Platform | Hydraulic Efficiency Improvement | Vehicle Efficiency Improvement | DLA-Energy LOW $0.61/Liter | DLA-Energy HIGH $1.04/Liter | In-Theatre LOW $3.73/Liter | In-Theatre HIGH $4.61/Liter |
|---|---|---|---|---|---|---|
| 10K Forklift | 2.70% | 2.70% | $4,460,990 | $7,530,304 | $26,937,516 | $33,247,719 |
| D6K Dozer | 2.70% | 2.70% | $122,629 | $207,002 | $740,489 | $913,952 |
| D7R-II Dozer | 2.00% | 2.00% | $988,607 | $1,668,802 | $5,969,663 | $7,368,076 |
| JD240 HYEX | 1.80% | 1.80% | $94,755 | $159,949 | $572,173 | $706,206 |
| RTCH | 2.00% | 1.40% | $507,162 | $856,106 | $3,062,475 | $3,779,869 |
| Total | | | $8,633,108 | $14,572,981 | $52,130,690 | $64,342,480 |

## 4. Conclusions

Two commercial energy efficient fluids were compared to military standard SAE 10 engine oil in a hydraulic pump and motor dynamometer. Three hydraulic motors were evaluated under low- and high-speed conditions at 50 °C and 80 °C. The EEHF reduced frictional torque losses in the hydraulic motors at low speeds. The EEHF also reduced the average flow losses by 3.6% to 6.4% at 80 °C. An analysis of the input and output power of the hydraulic dynamometer was conducted. The EEHF produced more torque with less input power, particularly under low-speed and high-pressure conditions. The potential fleet-wide fuel savings from EEHF was estimated. The savings ranged from $8,000,000 on a direct cost basis to $64,000,000 per year fully burdened. These benefits may be accrued in addition to the benefit of reducing the frequency of high-risk refueling operations and emissions.

The EEHF presented in this paper are used in a variety of commercial off-highway vehicles. These fluids are routinely selected for use in machines that operate at high intensity in harsh environments due to their premium performance capabilities. While the requirements of the US Army are unique, in terms of use in CE/MHE equipment, EEHF fluids are more suitable than SAE 10 oil in hydraulic applications due to superior water tolerance, filterability, air release, and efficiency. Other challenges, such as creating an objective definition of what constitutes an energy efficient hydraulic fluid and understanding the supply-chain ramifications will be the focus of future work.

**Author Contributions:** Conceptualization, J.B. and P.M.; methodology, P.M.; validation, E.S., J.B. and P.M.; formal analysis, P.M.; investigation, P.M.; resources, J.B. and E.S.; data curation, P.M.; writing—original draft preparation, P.M.; writing—review and editing, J.B. and E.S.; visualization, P.M.; supervision, E.S.; project administration, E.S.; funding acquisition, J.B. All authors have read and agreed to the published version of the manuscript.

**Funding:** This research received no external funding.

**Data Availability Statement:** DISTRIBUTION A. Approved for public release; distribution unlimited. OPSEC #: 6477. The data presented in this study are available on request from the corresponding author.

**Conflicts of Interest:** The authors declare no conflict of interest.

## Abbreviations/Nomenclature

| | | |
|---|---|---|
| ASABE | American Society of Agricultural and Biological Engineers | |
| BHL | Backhoe Loader | |
| *BMEP* | Brake Mean Effective Pressure | MPa |
| CE/MHE | Combat Engineering/Material Handling Equipment | |
| CI | Confidence Interval | |
| DEVCOM | U.S. Army Combat Capabilities Development Command | |

| | | |
|---|---|---|
| DL | Deployment Level | |
| DLA | Defense Logistics Agency | |
| DOD | Department of Defense | |
| EEHF | Energy Efficient Hydraulic Fluid | |
| EPA | Environmental Protection Agency | |
| *FMEP* | Friction Mean Effective Pressure | MPa |
| *FR* | Fuel Rate | Liters per hour |
| FY | Fiscal Year | |
| HD | Heavy Duty (work intensity) | |
| HYEX | Hydraulic Excavator | |
| ISO | International Organization for Standardization | |
| LD | Light Duty (work intensity) | |
| *LHV* | Lower Heating Value | kj/kg |
| LHS | Latin Hyperspace | |
| LPM | Liters per Minute | |
| MOVES | MOtor Vehicle Emission Simulator | |
| OEM | Original Equipment Manufacturer | |
| OE/HDO | Lubricating Oil, Internal Combustion Engine, Tactical | |
| OMS/MP | Operational Mission Statement/Mission Profile | |
| OPSEC | Operations Security | |
| PTO | Power Take Off | |
| QPLRPM | Military Qualified Product ListRevolutions Per Minute | |
| RTCH | Rough Terrain Cargo Handler | |
| SSL | Skid-steer Loader | |
| VG | Viscosity Grade | |
| $\alpha$ | | |
| $P_{ENG}$ | Engine Power | kW |
| $P_{HYD}$ | Hydraulic Power | kW |
| $N$ | Engine Speed | RPM |
| $\Delta p$ | Differential Pressure | Bar |
| $V$ | Engine Displacement | Liter per revolution |
| $\eta$ | Efficiency | |

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
