# Peer review of "Dynamometer Testing of Energy Efficient Hydraulic Fluids and Fuel Savings Analysis for US Army Construction and Material Handling Equipment"

_lubricants, doi:10.3390/lubricants10090216_

Round 1

Reviewer 1 Report

The paper investigates energy efficient hydraulic fluids in terms of reduction of the work-specific energy consumption of the combat engineering/material handling equipment (CE/MHE) in the US Army. This investigation includes a combination of dynamometer testing and fuel consumption models in order to estimate the potential equipment fuel savings for 11 US Army CE/MHE vehicle platforms. Two commercial energy efficient fluids have been compared to military standard SAE 10 engine oil. The comparisons include hydraulic motor torque losses, hydraulic system flow losses pump input and motor output power, efficiency and fuel consumption. The two commercial energy efficient fluids outperform SAE 10 engine oil in most of the searched parameters, an important outcome of the paper.

The authors are to be congratulated for their excellent contribution to lubrication engineering practice. I recommend the paper to be published in Lubricants – very nice work indeed.

These are two minor items that need to be considered in the final manuscript:

1. Lines 119 and 123: Please arrange reference numbers in consecutive order here and in the list of references.

2. Page 18: Please add missing acronyms in the table (EPA, FY, PTO for example).

Author Response

Dear Editor,

Thank you for your helpful comments, as well as the encourgaging words.  The suggested edits have been made.

Respectfully submitted,

Paul Michaep

Reviewer 2 Report

The authors should be congratulated on an excellent case study showing how improvements in hydraulic effiency, due to changing the lubricant, can result in fuel savings in off-road vehicles. Whilst the papers referenced by the authors from Holmberg and Erdemir have made a "first pass" attempt to calculate cost savings, papers such as this, which make a much more detailed cost saving analysis, are greatly needed by the tribological community for different application areas. I have a few comments below which may help the authors improve the paper still further. 

1. The authors mention, on page 2, that modern hydraulic oils perform better than the standard SAE 10W oil in terms of water tolerance, filterability and air release, in addition to efficiency. It is known that filterability and air in an oil can impact hydraulic losses. Are these additional benefits of the alternative hydraulic fluids included in the cost saving estimates?

2. It is not clear from the paper whether the EEHF-A and EEHF-B fluids contain viscosity modifiers (or whether the higher VI of these fluids simply comes from the type of base oil used). The SAE 10W oil should not contain a viscosity modifier as it is a monograde oil. It is slightly surprising, to me, that Table 2 shows a similar level for shear stability. Is this something the author could comment on?

3. Figure 3 shows significant hydraulic energy savings for the EEHF-A and EEHF-B fluids. Can the author comment on the size of the error bars on the data points shown in Figure 3?

4. At the bottom of page 9, the author comments that differences in energy saving performance could be due to the way different displacement motor designs respond to friction modifiers. The authors should state whether the different fluids, in Table 2, contain friction modifiers or not.

5. Some readers may be uneasy at seeing graphs (Figures 11-13) with normalized efficiencies greater than 100%. Perhaps a better way of explaining this is that the losses in the system are reduced when the EEHF-A and EEHF-B fluids are used. My understanding is that the efficiency will be less than 100% for all three fluids, but that the SAE 10W oil will have the lowest efficiency. 

6. I think Section 3.5, which focusses on fuel consumption, is a strong part of the paper. It would be useful if the author could comment on why there is such a large difference between estimated fuel consumption rates of the EPA and ASABE models.

7. The author should check that all the acronymns mentioned in the paper are included in the Acronyms/Symbols section (page 18). I noticed that the acronymns OMS/MP (page 14) and PTO (power take off, mentioned on page 13 were not included in the Acronyms/Symbols section on page 18. 

Author Response

August 23, 2022

Dear Reviewer,

We appreciate the professional and constructive comments you have provided. The comments have been addressed carefully in the revised version of the manuscript.  A response to the academic reviewer comments is provided below.

  1. The authors mention, on page 2, that modern hydraulic oils perform better than the standard SAE 10W oil in terms of water tolerance, filterability and air release, in addition to efficiency. It is known that filterability and air in an oil can impact hydraulic losses. Are these additional benefits of the alternative hydraulic fluids included in the cost saving estimates? The reviewer is correct that technical improvements in these areas can impact hydraulic losses. The cost savings estimate only included energy savings due to reductions in leakage flow and torque losses because the effects of improved water tolerance, air release, and filterability are more difficult to quantify.  
  2. It is not clear from the paper whether the EEHF-A and EEHF-B fluids contain viscosity modifiers (or whether the higher VI of these fluids simply comes from the type of base oil used). The SAE 10W oil should not contain a viscosity modifier as it is a monograde oil. It is slightly surprising, to me, that Table 2 shows a similar level for shear stability. Is this something the author could comment on? The following comment has been inserted on page 3 lines 99-104: All three fluids met the ISO 46VG specification and had a relatively high viscosity index. Polymethacrylate viscosity index improvers were used in the energy efficient fluid formulations. The high viscosity index of the SAE 10 engine oil was attributed to dispersant polymer additives that are commonly used in crankcase oil detergent-inhibitor packages.  All three oils had good shear-stability as measured in the ASTM D5621 Sonic Shear Test.
  3. Figure 3 shows significant hydraulic energy savings for the EEHF-A and EEHF-B fluids. Can the author comment on the size of the error bars on the data points shown in Figure 3? The following comment was added to the paragraph introducing figure 3. Since the torque loss measurements were highly repeatable, the confidence interval bars are very narrow and approximately the same size as the mean symbol.
  4. At the bottom of page 9, the author comments that differences in energy saving performance could be due to the way different displacement motor designs respond to friction modifiers. The authors should state whether the different fluids, in Table 2, contain friction modifiers or not. This is a very good comment. It is believed that friction modifiers are in Fluids A and B based upon data that is not approved for publication.  Two sections of the paper have been modified to clarify the proposed explanations for why the motors differed in their response to the EEHF fluids.

Page 8 lines 227 to 231 It is hypothesized that this is due to the higher displacement-to-weight ratio and lower operating speed of the radial piston motor. (See Table 3) This combination produces high contact pressures and low sliding speeds that shift tribological conditions toward the boundary lubrication regime where surface active additives can significantly impact friction.

Page 9 Lines 241 to 245 It is hypothesized that the metallurgy of the tribological elements in the variable displacement motor were relatively inert and less susceptible to the effects of surface-active additives in the fluid. 

  1. Some readers may be uneasy at seeing graphs (Figures 11-13) with normalized efficiencies greater EEHF-A and EEHF-B fluids are used. My understanding is that the efficiency will be less than 100% for all three fluids, but that the SAE 10W oil will have the lowest efficiency. 

This comment points out that a better explanation of Normalized Work-Specific Efficiency is required.  Page 10 Lines 249 to 256 have been revised to clarify its meaning. 

Normalized work-specific efficiency is useful for comparing fluids because hydraulic system performance is highly affected by operating conditions. It is determined by dividing the hydraulic system efficiency of the test fluid at a given operating speed, pressure, and temperature, by the hydraulic system efficiency of a baseline fluid under identical conditions. In this case, EEHF A and B are the test fluids, and SAE 10 is the baseline. Hydraulic system efficiency is defined as the ratio of motor output power to pump input power. When this ratio is higher for the EEHF than SAE 10, the normalized work-specific efficiency is greater than 1.0. 

  1. I think Section 3.5, which focusses on fuel consumption, is a strong part of the paper. It would be useful if the author could comment on why there is such a large difference between estimated fuel consumption rates of the EPA and ASABE models.

There is an explanation for the difference in the models.  Page 14 Lines 339-343 have been revised.  

As shown in Table 5, the fuel consumption rates for the EPA model were lower than the rates predicted by the ASABE model. This is because the ASABA model was developed for high tractive effort agricultural machines while the EPA model was developed for on-road vehicles.

  1. The author should check that all the acronyms mentioned in the paper are included in the Acronyms/Symbols section (page 18). I noticed that the acronyms OMS/MP (page 14) and PTO (power take off, mentioned on page 13 were not included in the Acronyms/Symbols section on page 18.

The paper has been carefully reviewed for missing acronyms and symbols.  Page 18 has been revised accordingly

Respectfully submitted,

Paul Michael